# Effectiveness of Breastfeeding Support Packages in Low- and Middle-Income Countries for Infants under Six Months: A Systematic Review

**DOI:** 10.3390/nu13020681

**Published:** 2021-02-20

**Authors:** Ritu Rana, Marie McGrath, Ekta Sharma, Paridhi Gupta, Marko Kerac

**Affiliations:** 1Indian Institute of Public Health Gandhinagar, Gujarat 382042, India; sharmase25@gmail.com (E.S.); drparidhigupta@gmail.com (P.G.); 2Department of Population Health, London School of Hygiene & Tropical Medicine, London WC1E 7HT, UK; marko.kerac@lshtm.ac.uk; 3GOAL Global, A96 C7W7 Dublin, Ireland; 4Emergency Nutrition Network, Oxford OX5 2DN, UK; marie@ennonline.net; 5Centre for Maternal, Adolescent, Reproductive & Child Health (MARCH), London School of Hygiene & Tropical Medicine, London WC1E 7HT, UK

**Keywords:** infant, breastfeeding, malnutrition, nutritionally at-risk, review

## Abstract

Small and nutritionally at-risk infants under six months, defined as those with wasting, underweight, or other forms of growth failure, are at high-risk of mortality and morbidity. The World Health Organisation 2013 guidelines on severe acute malnutrition highlight the need to effectively manage this vulnerable group, but programmatic challenges are widely reported. This review aims to inform future management strategies for small and nutritionally at-risk infants under six months in low- and middle-income countries (LMICs) by synthesising evidence on existing breastfeeding support packages for all infants under six months. We searched PubMed, CINAHL, Cochrane Library, EMBASE, and Global Health databases from inception to 18 July 2018. Intervention of interest were breastfeeding support packages. Studies reporting breastfeeding practices and/or caregivers’/healthcare staffs’ knowledge/skills/practices for infants under six months from LMICs were included. Study quality was assessed using NICE quality appraisal checklist for intervention studies. A narrative data synthesis using the Synthesis Without Meta-analysis (SWiM) reporting guideline was conducted and key features of successful programmes identified. Of 15,256 studies initially identified, 41 were eligible for inclusion. They were geographically diverse, representing 22 LMICs. Interventions were mainly targeted at mother–infant pairs and only 7% (*n* = 3) studies included at-risk infants. Studies were rated to be of good or adequate quality. Twenty studies focused on hospital-based interventions, another 20 on community-based and one study compared both. Among all interventions, breastfeeding counselling (*n* = 6) and education (*n* = 6) support packages showed the most positive effect on breastfeeding practices followed by breastfeeding training (*n* = 4), promotion (*n* = 4) and peer support (*n* = 3). Breastfeeding education support (*n* = 3) also improved caregivers’ knowledge/skills/practices. Identified breastfeeding support packages can serve as "primary prevention" interventions for all infants under six months in LMICs. For at-risk infants, these packages need to be adapted and formally tested in future studies. Future work should also examine impacts of breastfeeding support on anthropometry and morbidity outcomes. The review protocol was registered in the International Prospective Register of Systematic Reviews (PROSPERO 2018 CRD42018102795).

## 1. Introduction

Infant and child malnutrition in low- and middle-income countries (LMICs) is a major public health problem that requires urgent global attention [1]. Target 2.2 of the Sustainable Development Goals aims to “By 2030, end all forms of malnutrition, including achieving, by 2025, the internationally agreed targets on stunting and wasting in children under-five years of age” [2]. Recent estimates suggest that wasting alone threatens the lives of some 50.5 million (7.5%) under-five children globally [3]. These children have a 2.3 times higher risk of mortality compared with those without anthropometric deficits [4]. 

In recent years, the treatment of older children with severe malnutrition in LMICs has been revolutionised by a public-health-orientated model of care: Community-based Management of Acute Malnutrition (CMAM) [5]. It is the youngest who are especially vulnerable to morbidity and associated mortality: of 5.6 million under-five child deaths each year globally, 4.2 million (75%) deaths occur within the first year of life [6]. Infants aged under six months (infants < 6 m) are however often neglected in CMAM programmes [7]. However, some 8.5 million infants < 6 m worldwide are wasted [8]. Their treatment is often more challenging due to underling differences in feeding, physiology, development, and a wide variety of potential reason underlying their malnutrition [9]. Additional factors like low birth weight (LBW) and long-term effects of infant <6 m malnutrition necessitate special attention [10]. 

Whilst CMAM programmes for older children focus on outpatient-based care, current guidelines for malnourished infants < 6 m only describe inpatient-based treatments [5]. The high burden of malnutrition among infants < 6 m along with challenges and limitations of inpatient management indicate a need for this to change [11]. A move towards community-based management would bring more opportunities for providing effective care through community-based strategies for infant feeding and facilitate earlier and greater coverage of infants in need of support [10]. Reflecting this, the World Health Organisation (WHO) 2013 guidelines on severe malnutrition, for the first time, included specific recommendations for community-based management of malnourished infants < 6 m [12]. At the core of these guidelines is support for breastfeeding.

The WHO/United Nations Children’s Fund (UNICEF) global strategy for infant and young child feeding (IYCF) recommends early initiation of breastfeeding (EIBF)—within an hour of birth and exclusive breastfeeding (EBF) for the first six months of life [13]. Though prevalence of breastfeeding varies by infant age, with sometimes large variations both between and within countries, overall rates are poor, with only some 37% of infants < 6 m exclusively breastfed in LMICs [14]. 

Past reviews have examined interventions to promote breastfeeding but most of these focus on broad IYCF (<2 years) practices applied to the general infant population [15,16,17,18]. This review aims to address the evidence gap on how to best support breastfeeding in a subpopulation of small and nutritionally at-risk infants < 6 m, defined as those with wasting, underweight or other forms of growth failure [19,20]. Specific objectives include to:(1)identify and describe details of currently available breastfeeding support packages from LMICs for infants < 6 m,(2)assess the impact of breastfeeding support packages on breastfeeding practices, and(3)assess the impact of existing breastfeeding support packages on the knowledge/skills/practices of healthcare staff and caregivers.

## 2. Methods 

We developed and followed a standard systematic review protocol (PROSPERO 2018, CRD42018102795) in accordance with the PRISMA (preferred reporting items for systematic review and meta-analysis protocols) statement [21]. 

### 2.1. Search Strategy 

We conducted the search process in five databases—PubMed, CINAHL, Cochrane Library, EMBASE and Global Health. We used the following search strategy for PubMed: ((((“infant”[MeSH Terms] OR “infant”[tiab] OR “infants”[tiab]))) AND ((((((((((“education”[Subheading] OR “education”[tiab] OR “education”[MeSH Terms]))) OR ((“health education”[MeSH Terms] OR “health education”[tiab]))) OR ((“counselling”[tiab] OR “counseling”[MeSH Terms] OR “counseling”[tiab]))) OR ((train[tiab] OR “training”[tiab]))) OR ((session[tiab] OR sessions[tiab]))) OR support[tiab])) OR (((((“health personnel”[MeSH Terms] OR “health personnel”[tiab]))) OR ((“community health workers”[MeSH Terms] OR (“community”[tiab] AND “health”[tiab] AND “workers”[tiab]) OR “community health workers”[tiab] OR (“community”[tiab] AND “health”[tiab] AND “worker”[tiab]) OR “community health worker”[tiab]))) OR (((“caregivers”[MeSH Terms] OR “caregivers”[tiab] OR “caregiver”[tiab]) OR (“caregivers”[MeSH Terms] OR “caregivers”[tiab])))))) AND ((“breast feeding”[MeSH Terms] OR (“breast”[tiab] AND “feeding”[tiab]) OR “breast feeding”[tiab])). We used similar keywords with other selected databases. We limited the evidence to abstracts published in the English language from inception to 18 July 2018. 

### 2.2. Eligibility Criteria 

Population: We reviewed studies involving infants < 6 m. 

Intervention: Studies were eligible if they focused on breastfeeding support packages as the intervention of interest. Support packages were defined as—any form of breastfeeding education, training, counselling, and support provided to either healthcare staff and/or caregivers aimed to improve knowledge/skills/practices and/or breastfeeding practices. 

Outcome: Studies reporting on at least one of the following outcomes—breastfeeding practices, knowledge/skills/practices of healthcare staff and/or caregivers. 

Study design: We selected studies that included randomised control trials (RCTs), quasi-experimental, cohort, cross-sectional and other comparative observational studies. 

Context: We included studies that focused on LMICs [22] since this is the setting with the greatest burden of infant malnutrition globally. 

### 2.3. Study Selection 

All identified records were imported in Eppi Reviewer software (version V.4.7.1.1, EPPI-Centre, UCL Institute of Education, University of London, London, UK). One reviewer (RR) screened all titles and abstract. Two reviewers (RR and ES) independently screened the full text of studies that potentially met the inclusion criteria, any disagreements were resolved by a third reviewer (MK). 

### 2.4. Quality Assessment 

The National Institute for Health and Care Excellence (NICE) intervention studies checklist was used to assess the quality of included studies [23]. This checklist assesses study quality across five sections, Section 1 assess the external validity based on population characteristics; Section 2, Section 3 and Section 4 assess the internal validity based on—randomisation, allocation, intervention and control conditions, outcome assessed, and method of analysis; and Section 5 gives an overall grading for internal and external validity. The grading is given as good (++, all or most of the checklist criteria have been fulfilled, where they have not been fulfilled the conclusions are very unlikely to alter), adequate (+, some of the checklist criteria have been fulfilled, where they have not been fulfilled, or not adequately described, the conclusions are unlikely to alter), and poor (−, few or no checklist criteria have been fulfilled and the conclusions are likely or very likely to alter). Overall study quality is mentioned as internal and external validity grading. One reviewer first conducted quality assessment, which was then checked by a second reviewer. 

### 2.5. Data Extraction 

One reviewer (RR) extracted data using standard data extraction tool developed for this study. A second reviewer (PG) checked the extracted data in Eppi Reviewer. We extracted data on population (including sample size, details of setting such as hospital or community, country), intervention (description, type, delivery, and follow-up), comparison, outcome (description, type of measurement, effect size, and strength of evidence), and study design. In addition to three main outcomes, we also extracted data on two additional outcomes—morbidity and anthropometry.

### 2.6. Analysis and Reporting 

Since the review examines a diverse range of interventions and outcomes, meta-analyses was deemed inappropriate and thus, we opted to perform a narrative synthesis using the Synthesis Without Meta-analysis (SWiM) reporting guideline [24]. The studies were grouped by intervention categories as follows: counselling, education, training, promotion, peer support, and others. Outcomes were grouped in five categories: breastfeeding practices, caregivers’ knowledge/skills/practices, healthcare staff knowledge/skills/practices, morbidity, and anthropometry. Many of the studies reported several measures of the same outcome and/or measured outcomes at different time points, resulting in inconsistency in the effect measures. The findings were first summarised by outcome in tables—we included differences (mean/median/prevalence) and/or ratio (prevalence/risk/hazard) measures between intervention (IG) and control group (CG), as reported by the individual studies. Thereafter, we summarised the findings by transforming difference/ratio measures to standardised metric-direction of effect (positive/negative/mixed/no effect). Lastly, to find out "is there any evidence of an effect" of five intervention categories on each outcome, the evidence was synthesised in tabular form by vote counting based on direction of effects [25]. We calculated proportion of effects if for each outcome and intervention category there were three or more comparisons available. 

### 2.7. Ethics Approval

We submitted the review protocol to the Research Ethics Committee at London School of Hygiene and Tropical Medicine (LSHTM MSc Ethics ref: 15968). Being a systematic review of publicly available literature the review was assessed and judged as not requiring ethical approval. 

## 3. Results

Figure 1 presents the selection process and search results. The search identified 15,256 records. After duplicate removal and screening titles and abstracts, 155 records were eligible for full-text review—of these 16 were available in "abstract only" form. Of the remaining 139 records, 98 did not meet the inclusion criteria. A final 41 studies were included in the main analysis. 

### 3.1. General Characteristics of the Included Studies

Table 1 summarises key characteristics of the included studies. They were geographically diverse, representing 22 LMICs. Most were either RCTs (RCTs, *n* = 15, 36.6% and Cluster-RCTs, *n* = 9, 21.9%) or quasi-experimental (*n* = 14, 34.1%) studies. The number of individual participants studied ranged from 60 to 2579. In eleven studies (27%), the intervention was aimed at mother–infant pairs. Eight studies (20%) also targeted fathers of infants < 6 m, three studies (7%) included small/at-risk infants, while one study (2%) each targeted adolescent mothers and healthcare staff. 

Studies were generally rated to be of good or adequate quality (Table 1). Overall, seven studies were rated good quality (++,++), 15 adequate (+,+), and 10 poor (−,−) for both internal and external validity. Four studies were rated adequate quality for internal validity and poor quality for external validity (+,−). Another four studies were rated poor quality for internal validity and adequate quality for external validity (−,+). One study was rated as adequate for internal and good for external validity (+,++). 

Of the 41 studies, 20 (49%) focused on hospital-based interventions, another 20 (49%) on community-based, and one study compared both (Table 1). Of 20 (49%) community-based studies, three (7%) were multi-country. Among hospital-based, seven assessed education interventions, five counselling, four promotion, three training, and one focused on ten steps of successful breastfeeding. Of community-based, six assessed counselling interventions, five training, four peer support, two education, two promotion, and one focused on large-scale multi-component intervention. Of the total 41 studies, 37 (90%) measured breastfeeding outcomes, 12 (29%) caregivers’ knowledge/skills/practices, and two (5%) healthcare staffs’ knowledge/skills/practices. Additionally, nine studies (22%) also reported morbidity and another eleven (27%) reported outcomes on anthropometry. None of the studies reported on mortality following interventions. 

### 3.2. Breastfeeding Interventions and Their Effect on Various Outcomes

A summary of direction of effect and proportion of effects of breastfeeding interventions on various outcomes is presented in Table 2 and Table 3, respectively. A more detailed summary (intervention components and outcome measures) is presented in Appendix A. The subsequent section briefly describes the effect and key features of included studies.

#### 3.2.1. Breastfeeding Practices 

Of the 37 studies reporting breastfeeding practices, three sets of studies reported similar data and nine studies reported more than one comparisons, making up a total of 42 comparisons for synthesis. Of these 42 effects, 12 assessed counselling, 10 education, nine training, five promotion, two peer support, and four comparisons assessed other type of interventions. 

Counselling: Of 12 counselling category effects, seven effects favour interventions (58%, 95% CI 31% to 80%, *p* = 0.77). Five studies (12%) focused on hospital-based counselling. Three compared counselling with standard care. One study, which included premature infants, reported positive effect in EBF from birth to four months, while other two studies, which included normal infants, found no effect on EBF at 4/6 months. Another study compared pre-, peri-, and post-natal counselling (IG1) and peri- and post-natal counselling (IG2) with standard care (CG). Authors reported positive effect with both intervention groups in continuation of EBF from 1 to 6 months. One study compared four groups, adolescent mothers alone (with and without intervention) and with their mothers (with and without intervention), they reported positive effect in EBF in both intervention groups. Six studies (15%) examined effect of community-based counselling. Two studies reported positive effect in EBF at six months in Burkina Faso, Uganda, and South Africa, while another two from Kenya and one from Mexico reported no effect. A sixth study from Brazil reported positive effect in EBF at four months.

Education: Of 10 education category effects, eight effects favour interventions (80%, 95% CI 49% to 94%, *p* = 0.10). Six studies (15%) assessed the effect of hospital-based education. Three studies compared educational interventions with standard care. Of these, two studies reported positive effect in EBF at four months, while one reported no effect on EBF at six months. Another three studies assessed the effect of educating fathers (IG1: mother and father, IG2: mothers only). A study from Brazil reported a positive effect on EBF at six months among mother and father group compared to no intervention group, while a comparison between educating mothers only with no intervention group showed no effect. Two other studies reported a positive effect on EBF at six months compared to mothers only/no intervention group. Two studies (5%) assessed the effect of community-based education. One study reported positive effect on EBF at six months and duration of EBF, while other study reported positive effect in delaying time of introduction of water from 4 to 6 months.

Training: Of the nine training category effects, five favour interventions (55%, 95% CI 26% to 81%, *p* = 1.00). Two studies (5%) focused on hospital-based training. One study reported positive effect in EBF from 2 weeks to 6 months as a result of breastfeeding (IG1) and child feeding (IG2) counselling by trained counsellors, while other study that assessed the effect of training movies reported no effect on EBF at six months. Five studies (12%) focused on community-based training. Three studies reported positive effect in EBF at six months. One study compared effect of visits by specially trained CHWs with visits by normally trained CHWs and found no effect on EBF at six months. Another study compared effect of training traditional birth attendants (TBAs)/community volunteers (CVs) and training + supervision of TBAs/CVs with standard care and found a mixed effect (an improvement in EIBF, however no effect was observed in EBF at six months).

Peer support: Of the two peer support effects, only one favours intervention. Four studies (8%) assessed effect of community-based peer support. Three studies, by a similar research group and reporting similar data, compared effect of training fathers with no training to fathers; authors reported positive effect in EBF at six months in intervention group. Another study assessed the effect of mother-to-mother support programme. This study found a mixed effect on breastfeeding practices.

Promotion: Of the five promotion category effects, all five effects favour interventions (100%, 95% CI 56% to 100%, *p* = 0.06). Three studies (7%) assessed the effect of hospital-based promotion. One study evaluated the effect of breastfeeding motivation programme and the other was based on the theory of planned behaviour. The study on motivation programme reported positive effect in the proportion of mothers with the intention to breastfeed. Other study on theory of planned behaviour found positive effect in EBF at six months. Another study compared promotion through peer support group (IG1) and healthcare providers’ education (IG2) with routine care; this study found positive effect in mean duration of EBF at two months in both intervention groups. One study evaluated the effect of community-based promotion through health messages. These messages focused on advantages of breastfeeding and disadvantages of bottle-feeding. Authors reported positive effect on breastfeeding practices. 

Other interventions: One hospital-based study compared the effect of baby friendly hospital initiative (BFHI) steps 1–9 and BFHI steps 1–10 with standard care. Authors reported no effect on EIBF with either intervention. Another community-based study assessed the effect of intensified interpersonal counselling (IPC), mass media (MM), community mobilisation (CM) and policy advocacy (PA) with standard nutrition counselling and less intensive IPC, MM, CM, and PA in Bangladesh and Vietnam; authors reported an overall positive effect in EIBF within 1 h and EBF at six months in both countries. One study compared the effect of home-based intensive counselling (HBIC) and facility-based semi-intensive counselling (FBSIC) with standard care. This study found, positive effect of HBIC on EBF at six months, while there was no effect of FBSIC when compared with standard care. 

#### 3.2.2. Caregivers’ Knowledge/Skills/Practices 

Of the 12 studies reporting caregivers’ knowledge/skills/practices, three studies reported similar data, leaving 10 comparisons to contribute to the synthesis. Of these 10 effects, one assessed counselling, four assessed education and another four assessed promotion. 

Counselling: One study assessed the effect of hospital-based counselling using BASNEF (Beliefs, Attitudes, Subjective Norms, and Enabling Factors) model. Authors reported positive effect in both mean scores of structures in BASNEF model and mean lactation performance scores. 

Education: Of the four education category effects, three effects favour interventions (75%, 95% CI 30% to 95%, *p* = 0.62). Three studies (7%) focused on hospital-based educational interventions. One study compared the effect of breastfeeding education to mother and father with education to mothers only. This study reported positive effect in both mean breastfeeding knowledge and mean breastfeeding attitude scores. Another study compared educational intervention based on PRECEDE (Predisposing, Reinforcing, Enabling Constructs in Educational Diagnosis and Evaluation) model with standard care. Authors observed positive effect in mean knowledge scores. A third study with one-to-one education session found a mixed effect on breastfeeding knowledge. One study examined the effect of a community-based educational intervention based on the 18 h WHO/UNICEF breastfeeding counselling/lactation management course. Authors reported positive effect in mean knowledge scores at six weeks.

Promotion: Of the four promotion category effects, two effects favour interventions (50%, 95% CI 15% to 84%, *p* = 1.00). Two studies (5%) assessed the effect of hospital-based breastfeeding promotion on self-efficacy. One study reported positive effect in mean self-efficacy scores, while other observed no effect. Two studies (5%) focused on community-based breastfeeding promotion. One study that assessed the effect of an audio programme observed a mixed effect on caregivers’ knowledge/skills/practices, while another study that assessed the effect of health messages reported positive effect in knowledge of advantage of colostrum and EIBF. 

Peer support: Three studies (7%), by similar research group and reporting similar data, reported the effect of community-based education on fathers and observed positive effect in mean knowledge, attitude and total scores. 

#### 3.2.3. Healthcare Staff Knowledge/Skills/Practices 

Of two studies reporting on the knowledge/skills/practices of healthcare staff, one study reported more than one comparison, making up to three comparisons to contribute to the synthesis. All three assessed training interventions. Of the three training category effects, one effect favours intervention (33%, 95% CI 6% to 79%, *p* = 1.00). One study evaluated the effect of hospital-based breastfeeding DVD training. Authors reported positive effect in both mean knowledge and confidence scores of healthcare professionals compared to the control group. Another study that evaluated the effect of community-based special training on TBAs/CVs observed a mixed effect—authors found a positive effect of training on TBAs/CVs knowledge on EIBF, but no effect on other knowledge items. 

#### 3.2.4. Morbidity 

Of the nine studies reporting morbidity, two studies reported similar data and one study reported more than one comparison, making up to total nine comparisons for synthesis. Of these nine effects, three assessed counselling, another three assessed education, two training, and one comparison assessed other type of intervention. 

Counselling: Of the three counselling category effects, two effects favour interventions (66%, 95% CI 20% to 93%, *p* = 1.00). Three studies (7%) focused on community-based counselling. Two studies, reporting similar data, reported no effect of counselling on diarrheal morbidity in Burkina Faso, Uganda, and South Africa. Another study from Mexico that assessed the effect of home-based peer counselling with six visits (IG1) and three visits (IG2) reported positive effect on diarrhea among 0–3 months in both intervention groups. 

Education: Of the three education category effects, none favours intervention. Two studies (5%) assessed the effect of hospital-based education programmes. One study reported no effect on mild illness and hospitalisation at six months. Similarly, another study also reported no effect on diarrheal and respiratory illness. One study on the effect of community-based health education reported no effect on diarrheal morbidity from 1 week to 6 months. 

Training: One study evaluated the effect of hospital-based training movies. These included the importance of breastfeeding and ways of doing it. Authors reported no effect on infant morbidity at six months. Another study evaluated the effect of community-based training for existing primary healthcare workers and reported mixed results for diarrheal morbidity. 

Other interventions: One study that compared the effect of BFHI steps 1–9 and BFHI steps 1–10 with standard care, reported no effect on fever with cough at six months.

#### 3.2.5. Anthropometry 

Of the 11 studies reporting anthropometry, two studies reported similar data, leaving 10 comparisons for synthesis. Of these 10 effects, three assessed counselling, another three assessed education, four training, and one promotion intervention. 

Counselling: Of the three counselling category effects, only one effect favours intervention (33%, 95% CI 6% to 79%, *p* = 1.00). Two studies (5%) examined the effect of hospital-based counselling. One study, with BASNEF model based counselling, reported positive effect in mean infant weight from birth to 4 months. The other study did not find any effect of counselling on mean weight. Two studies (5%), reporting similar data, assessed the effect of community-based counselling in Burkina Faso, Uganda, and South Africa. Overall, authors observed no effect on weight-for-length z score (WLZ), length-for-age z score (LAZ), weight-for-age z-score (WAZ), wasting, stunting, and underweight at six months. These two studies also reported breastfeeding outcomes. Both studies found a positive effect on breastfeeding practices but no effect was found on anthropometric outcomes. 

Education: One study focused on hospital-based education intervention and found positive effects on mean weight and length of infants at four months. Another study assessed the effect of community-based health education and found no effect on median weight and WAZ.

Training: Of the four training category effects, none favours intervention. One study assessed the effect of hospital-based training movie; authors reported no effect on mean weight from birth to 6 months. Three studies (7%) reported the effect of community-based training. Of these, two studies did not find any effect on mean weight, length, WAZ, and height-for-age z score (HAZ). The third study reported mixed effect. 

Promotion: One study assessed effect of hospital-based promotion on anthropometry and found no effect. Further details of the hospital and community-based interventions that showed a positive effect are presented in Appendix A. 

## 4. Discussion

### 4.1. Summary of Key Findings

Our systematic review highlights a number of key findings. First, although a good number of and variety of studies include infants aged <6 m, few are directly and exclusively targeted at small and at-risk infants in this age category. Second there are five broad categories of interventions: (1) counselling, (2) education, (3) training, (4) promotion, and (5) peer support showed evidence of positive effect on breastfeeding practices. Third, few studies evaluated the effect on caregivers’ knowledge/skills/practices. Fourth, evidence on training for healthcare staffs’ knowledge/skill/practices was weakest; fifth, few studies reported on anthropometric or morbidity, where these were reported, the impact was minimal (though we acknowledge that some papers on these outcomes may have been missed due to our search strategy); sixth, the intervention categories had many similarities and overlap between them; and seventh, key characteristics of interventions, outcomes and population were identified.

### 4.2. This Review’s Findings in Context

Although the interventions were grouped into categories (as defined by the authors) as counselling, education, training, promotion and peer support, there were many similarities and overlaps between them. For instance, interventions with a training programme for counsellors (training–counselling) [56], educational sessions by trained staff (education–training) [45], breastfeeding promotion using peer support (promotion–peer support) [49] and educating fathers as supporters (education–peer support) [52,53,54]. Because of these, it was important to look into details of what exactly each intervention involves. There is also potential future need for standardising definitions so that differences between these are clear. For example, “education” and “training”: whilst anyone can attend a training course, education might imply that participants have achieved some level of formal knowledge/skills standards as a result of the training. 

Generally, interventions with positive effect were those which: (1) were well structured (BASNEF model and PRECEDE model) [26,37], (2) delivered more frequently (reinforcing) [47,49,56], and (3) involved fathers into care as peer supporters [52,53,54]. Interventions with telephone lactation counselling and education sessions followed by follow-up with phone calls did not show the aimed-for effect. Interventions with mixed effect were those that used—(1) short training movies [59,60], (2) WHO/UNICEF training material [28,38], and (3) existing community health workers (CHWs) [32,62]. 

Most studies showed improvement in breastfeeding outcomes, most commonly measured as EBF prevalence. However, these outcomes were self-reported and not objectively measured [31,58,64]. Reporting of breastfeeding practices could be affected by many factors including social desirability bias, which could have led to the over-reporting of positive practices [64]. Future studies could consider objective measures, for example, assessment of milk output using an isotope dilution technique [67]. 

A particular challenge given the target population who inspired this review was that very few studies (n = 3) directly addressed or presented separate data for infants < 6 m who were already small, malnourished, or had established growth failure [26,37,56]. We included the wider group because lessons from the general population can be indirectly applied; we also hoped to identify databases with potential for future secondary analysis of subpopulations. 

Previous reviews reported interventions could improve breastfeeding outcomes [16,68,69,70,71]. Sinha et al. found counselling by healthcare staff and peers as key interventions to improve breastfeeding outcomes [16]. The group emphasised on delivering interventions in a combination of settings—health system, home and community—for achieving a higher impact. Haroon et al. conducted a review on the effect of breastfeeding promotion interventions on breastfeeding practices and found an increase in EBF rates with educational interventions [68]. Similar findings were reported by another review [69], where authors found a positive effect of breastfeeding education and/or additional support to mothers through counsellors on EBF rates. Shakya et al. (2017) reported community-based peer support for mothers as an effective intervention to improve EBF in LMICs [70]. Similar findings were also reported by other review where authors found peer support significantly decreased the risk of discontinuing EBF as compared to controls (RR: 0.71; 95% CI: 0.61–0.82) [71]. A 2016 review identified healthcare providers as key players in education and encouraging mothers to breastfeed [15]. Another review reported training of hospital healthcare staff effective in improving their knowledge/skills/practices [18]. Lately, a review reported quality evidence gap in breastfeeding education and training for healthcare staff [72]. 

### 4.3. Limitations and Strengths 

Review findings should be interpreted in light of the following limitations. First, selection bias could have occurred as only a single reviewer screened articles for inclusion. Despite this, a good number of articles were identified, so even if some others might have been missed, we feel it unlikely that the overall conclusions or messages arising from our review would be different. In addition, despite contacting corresponding authors three full-text articles could not be retrieved [73,74,75]. Second, only three studies focused directly on at-risk infants, the rest excluded both at-risk infants and mothers. Third, within included studies, there was significant heterogeneity for both interventions and outcomes. Although this was expected, given the scope of the review was broad and hence, a narrative synthesis is presented. Our synthesis addresses the question "is there any evidence of an effect" and does not quantify the average intervention effect. It does not account for difference in the relative sizes of the studies. As total number of studies contributing to the analysis was small, large uncertainty in the estimated proportion was expected. Fourth, a qualitative tool was used for quality appraisal, which could have introduced reviewer bias. Fifth, grey literature was not searched, this may have yielded additional studies but given the many limitations of those published in peer-review; is unlikely to have added any critical extra information. Another limitation is that the review did not look directly into interventions for infants who are not breastfed. Hence, the findings from this study are not generalisable to the entire infant <6 m population at-risk of, or with, growth failure.

Finally, we found few studies that reported on EBF-associated anthropometric change and morbidity. Where this was reported, the effect was mostly absent or limited. It is important to note however, that our search terms did not include anthropometry and morbidity, and thus some relevant studies might have been missed (we hypothesise that these would be few if any due to breastfeeding-outcomes being on the causal pathway to anthropometric/morbidity changes—but this should be explored in future work). This does not mean that improved EBF practices have no benefits, what matters is not anthropometric change alone but associated morbidity and mortality. These are difficult to evaluate since large sample sizes are needed; also much longer timeframes—given evidence about the benefits of breastfeeding on later life non-communicable diseases [2]. It does, however, makes it difficult to know what public health impact the described interventions would have if scaled up. Despite these limitations, this review also has notable strengths. To our knowledge, this is the first systematic review to provide evidence on breastfeeding support packages for infants < 6 m. Results can thus be used, with cautious generalisation, to inform future research and policy/practice in LMICs with a high burden of infant <6 m malnutrition. 

### 4.4. Implications for Practice and Research 

This review identified several breastfeeding support packages that can be applied to infants < 6 m in LMICs. Details about these support packages are presented in Appendix A. These interventions can directly serve as "primary prevention" interventions (IYCF) for all infants < 6 m in LMICs. However, for already small and at-risk infants, breastfeeding support packages need to be tailored and formally tested. Subsequently, these can be applied as "secondary/tertiary" interventions for infants who are at-risk of growth failure. Of note, these models can only be applied to infants who are breastfed. For those who are not currently breastfed due to various reasons [76] other interventions, such as—supplementary suckling, establishing/re-establishing breastfeeding, wet nursing, breastmilk substitutes, etc. should be explored [12]. These interventions will further add to the necessary support package for infants at risk of or with growth failure. 

Although updated (2013) WHO guidelines on the management of severe acute malnutrition (SAM) introduces community-based care for infants < 6 m with “uncomplicated SAM”, almost all LMICs are still following the inpatient care guidelines [10]. The community-based interventions identified in this review could be helpful for LMICs to develop context-specific national guidelines [77]. Engaging fathers as peer supporters, training local peer counsellors and home-based counselling by trained counsellors can be considered as strategies to support community-based management of at-risk infants [52,53,54]. 

This review also helped highlight some research gaps, including

(1)Which factors are associated with morbidity (diarrhea, respiratory illness, and hospitalisation) and growth failure among infants < 6 m who are exclusively breastfed.(2)Which breastfeeding interventions are effective in improving breastfeeding practices and associated morbidity/mortality outcomes for particularly at-risk infants (premature babies, preterm, LWB, twin babies, or babies with anthropometric deficits/growth failure).(3)Which breastfeeding training interventions are most effective in improving the healthcare staffs’ knowledge/skills/practices, (and what are essential features/characteristics of those interventions).(4)Which breastfeeding support packages are most valued by mothers/carers and why (e.g., maternal perceptions of value might also focus on time needed to attend; confidence gained during engagement; perception of being well supported by a particular intervention).

## 5. Conclusions

There is little data to directly inform the future feeding management of small and nutritionally at-risk infants < 6 m. However, many lessons from interventions focused on the wider group of infants and infants < 6 m can be usefully applied. The identified packages on breastfeeding counselling, education, training, promotion, and peer support showed evidence of positive effect on breastfeeding outcomes. The packages identified in our review can serve as "primary prevention" interventions for all infants < 6 m. It is likely that the same packages, with minimal adaptations, can also support small/at-risk infants < 6 m but their effectiveness for this population needs to be formally tested. In such future studies, it is important to focus on clinically important outcomes like improved anthropometry and reduced morbidity/mortality, not just self-reported outcomes like improved breastfeeding practices. 

## Figures and Tables

**Figure 1 nutrients-13-00681-f001:**
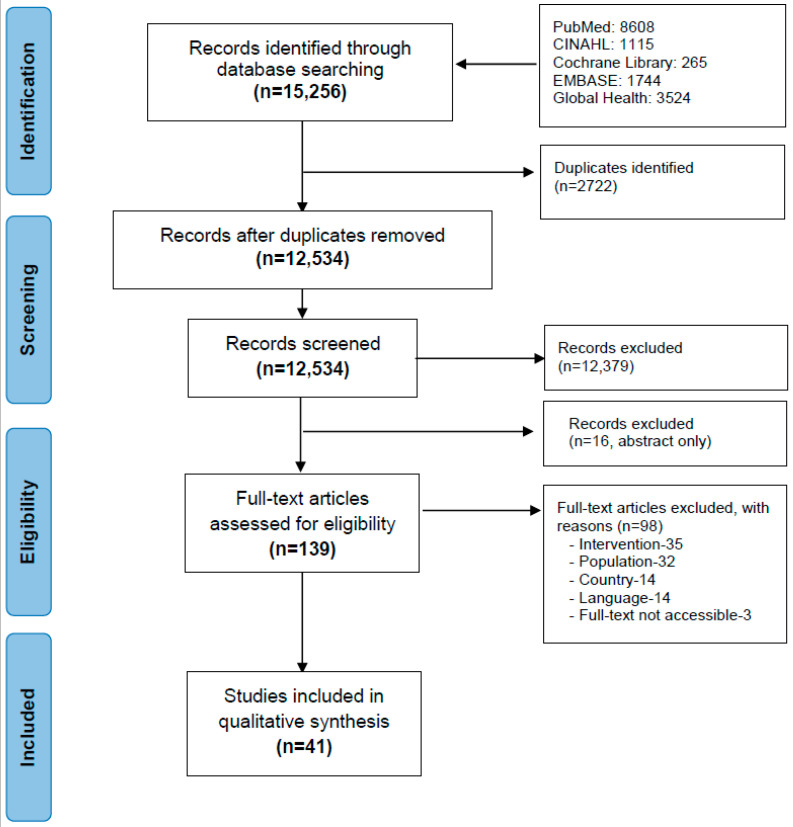
Flow diagram of included and excluded studies.

**Table 1 nutrients-13-00681-t001:** Key characteristics of included studies, n = 41.

Author (Year)	Country	Design	Quality (Internal, External Validity)	Population	Sample Size	Intervention Type	Outcomes				
							Breast-Feeding Practices	Care-Giver K/S/P	Health-Care Staff K/S/P	Morbidity	Anthropo-metry
**Counselling interventions**											
Ahmadi (2016) [26]	Iran	RCT	(+,++)	Mother–premature infants	124	Hospital-based	√	√			√
Aidam (2005) [27]	Ghana	RCT	(+,+)	Pregnant women	136	Hospital-based	√				
Albernaz (2003) [28]	Brazil	RCT	(+,+)	Infants	188	Hospital-based	√				√
Oliveira (2014) [29]	Brazil	RCT	(+,+)	AM–infants–grandmother	323	Hospital-based	√				
Engebretsen (2014) [30] Tylleskar (2011) [31] ^†^	Burkina Faso, Uganda, South Africa	C-RCT	(++,++)	Mother–infant pairs	2579	Community-based	√			√	√
Kimani-Murage (2017, 2016) [32,33] ^†^	Kenya	C-RCT	(++,++)	Mother–child pairs	1110	Community-based	√				
Leite (2005) [34]	Brazil	RCT	(+,−)	Mothers–infants	1001	Community-based	√				
Morrow (1999) [35]	Mexico	RCT	(−,−)	Pregnant women	130	Community-based	√			√	
Tahir (2013) [36]	Malaysia	RCT	(−,+)	Mothers	357	Hospital-based	√				
**Education interventions**											
Ahmed (2008) [37]	Egypt	QE	(−,−)	Mother–preterm infants	60	Hospital-based		√			
Aksu (2011) [38]	Turkey	RCT	(+,+)	Pregnant women	66	Community-based	√	√			
Froozani (1999) [39]	Iran	QE	(−,+)	Mother–infant pairs	120	Hospital-based	√			√	√
Jakobsen (2008) [40]	Guinea-Bissau	RCT	(+,+)	Mothers and infants	1721	Community-based	√			√	√
Khresheh (2011) [41]	Jordan	RCT	(+,+)	Primiparous women	140	Hospital-based	√	√		√	
Neyzi (1991) [42]	Turkey	QE	(−,−)	Mother–infant pairs	146	Hospital-based	√				
Ozluses (2014) [43]	Turkey	QE	(−,−)	Couples with their infants	117	Hospital-based	√				
Su (2016) [44]	China	QE	(−,−)	Pregnant women–partners	72	Hospital-based	√	√			
Susin (2008) [45]	Brazil	QE	(+,+)	Mother–father–infant triads	586	Hospital-based	√				
**Promotion interventions**											
Akram (1997) [46]	Pakistan	QE	(−,−)	Pregnant women	120	Community-based	√	√			
Cangol (2017) [47]	Turkey	RCT	(−,+)	Pregnant women	100	Hospital-based	√	√			√
Gu (2016) [48]	China	RCT	(−,+)	Pregnant women–husband/mother	352	Hospital-based	√				
Moudi (2016) [49]	Iran	QE	(+,−)	Pregnant women	108	Hospital-based	√				
Reinsma (2016) [50]	Cameroon	QE	(+,−)	Pregnant women–partners	384	Community-based		√			
Saljughi (2016) [51]	Iran	QE	(−,−)	Pregnant women	74	Hospital-based		√			
**Peer support interventions**											
Bich (2017, 2017, 2014) [52,53,54] ^†^	Vietnam	QE	(+,+)	Fathers–pregnant wives	492	Community-based	√	√			
Dearden (2002) [55]	Guatemala	B-A	(+,+)	Mothers of infants < 6 m	768	Community-based	√				
**Training interventions**											
Agrasada (2005) [56]	Philippines	RCT	(+,−)	Mother–infant (term LBW)	204	Hospital-based	√				
Balaluka (2012) [57]	Congo	PE	(+,+)	Infants	386	Community-based	√				√
Bhandari (2001) [58]	India	C-RCT	(++,++)	Infants	1115	Community-based	√			√	√
Khayyati (2009) [59]	Iran	RCT	(−,−)	Pregnant women	244	Hospital-based	√			√	√
Ma (2018) [60]	China	RCT	(+,+)	Healthcare staff	210	Hospital-based			√		
Mukantwali (2006) [61]	South Africa	QE	(−,−)	Mother–infant pairs	182	Community-based	√				
Mukhopadhyay (2017) [62]	India	C-RCT	(−,−)	Mother–infant pairs	130	Community-based	√				√
Shamim (2017) [63]	Bangladesh	C-RCT	(+,+)	Mothers of infants < 6 m	1182	Community-based	√		√		
**Other intervention**											
Menon (2016) [64]	Bangladesh, Viet Nam	C-RE	(++,++)	Infants	2000	Community-based	√				
Yotebieng (2015) [65]	Congo	C-RCT	(++,++)	Mother–infant pairs	975	Hospital-based	√			√	
Ochola (2013) [66]	Kenya	C-RCT	(+,+)	Pregnant women	360	Community-, Hospital-based	√				

Symbol: † indicates linked studies. Abbreviations: AM, adolescent mother; B-A, before-after; C-, cluster; K/S/P, knowledge/skills/practices; LBW, low birth weight; m, month; PE, program evaluation; QE, quasi-experimental; RCT, randomised controlled trial; RE, randomised evaluation.

**Table 2 nutrients-13-00681-t002:** Summary of effect of breastfeeding interventions on breastfeeding practices, knowledge/skills/practices of caregivers and healthcare staff, morbidity, and anthropometry outcomes.

Author, Population	Intervention and Comparison	Effect on Outcomes				
		Breastfeeding Practices	Caregiver K/S/P	Healthcare Staff K/S/P	Morbidity	Anthropo-Metry
**Counselling interventions**						
Ahmadi (2016) [26] Mothers with premature (34–37w) infants	IG: breastfeeding consultation sessions based on BASNEF model vs. CG: conventional training by staff	↑	↑			↑
Albernaz (2003) [28] Infants	IG: lactation counselling support by trained nurse- hospital and home visits vs. CG: standard care	↔				↔
Tahir (2013) [36] Mothers	IG: telephone lactation counselling twice monthly by certified lactation counsellors + conventional care vs. CG: conventional care	↔				
Aidam (2005) [27] ^‡^ Pregnant women attending prenatal clinics	IG1: lactation counselling- pre-, peri-, and post-natally vs. CG: non-breastfed health education support	↑				
	IG2: lactation counselling- peri-, and post-natally vs. CG: non-breastfed health education support	↑				
Oliveira (2014) [29] ^‡^ Adolescent mothers with newborn and their mothers	IG1: counselling sessions for adolescent girls vs. CG1: adolescent girls without intervention (not living with mother)	↑				
	IG2: counselling sessions for adolescent girls vs. CG2: adolescent girls without intervention (living with mother)	↑				
Kimani-Murage (2017, 2016) [32,33] ^†^ Mother–child pairs	IG: home-based nutrition counselling by CHWs (trained to offer counselling on MIYCN) vs. CG: standard care	↔				
Engebretsen (2014) [30] Tylleskar (2011) [31] ^†^ Mother–infant pairs	IG: peer counselling (1 antenatal and 4 postnatal) vs. CG: usual care	↑			↔	↔
Morrow (1999) [35] ^‡^ Pregnant women	IG1: home based peer counselling—6 visits vs. CG: no intervention	↔			↑	
	IG2: home based peer counselling—3 visits vs. CG: no intervention	↔			↑	
Leite (2005) [34] Mothers–infants	IG: home based peer counselling with home visits 5,15,30,60,90, and 120 days vs. CG: standard care	↑				
**Education interventions**						
Froozani (1999) [39] Mother–infant pairs	IG: education, face-to-face, after delivery and during follow-up vs. CG: usual care	↑			↔	↑
Neyzi (1991) [42] Mother–infant pairs	IG: 2 educational sessions after delivery vs. CG: usual care	↑				
Khresheh (2011) [41] Primiparous women	IG: one-to-one postnatal education sessions and follow-up phone calls at 2 and 4 months vs. CG: routine care	↔	↕		↔	
Susin (2008) [45] ^‡^ Mother–father–infant triads	IG1: educational session by a trained pediatrician to mother + father vs. CG: no intervention	↑				
	IG2: Educational session by a trained pediatrician to mother only vs. CG: no intervention	↔				
Ozluses (2014) [43] ^‡^ Couples with their infants	IG1: educating mothers + fathers—20 min/day vs. CG: no education	↑				
	IG2: educating mothers—20 min/day vs. CG: no education	↑				
Su (2016) [44] Pregnant women	IG: education to mother + father vs. CG: education to mother only	↑	↑			
Ahmed (2008) [37] Mothers and preterm infants (born <37w)	IG: 5 session (PRECEDE model) vs. CG: routine care		↑			
Aksu (2011) [38] Pregnant women	IG: BF education at home on day 3 postpartum (reinforcement) vs. CG: no education/support	↑	↑			
Jakobsen (2008) [40] Mothers and infants	IG: education provided individually and orally in local language vs. CG: standard care	↑			↔	↔
**Training interventions**						
Agrasada (2005) [56] ^‡^ Mother–infant (term LBW) pairs	IG1: BF counselling by trained counsellors vs. CG: any counselling	↑				
	IG2: child feeding counselling by trained counsellors vs. CG: any counselling	↑				
Khayyati (2009) [59] Pregnant women	IG: training movies and common method of face-to-face training vs. CG: face-to-face training	↔			↔	↔
Ma (2018) [60] Healthcare professional–doctor, nurse, midwife	IG: BF essential support skills DVD vs. CG: vaginal delivery DVD			↑		
Bhandari (2001) [58] Infants	IG: promotion by CHWs trained in BF (3-day course) vs. CG: usual care	↑			↕	↔
Balaluka (2012) [57] Infants	IG: trained CVs promoting EBF via door-to-door visits and community meetings vs. CG: usual care only	↑				↔
Mukantwali (2006) [61] Mother–infant pairs	IG: visited by specially trained CHW vs. CG: visited by normally trained CHW	↔				
Mukhopadhyay (2017) [62] Mother–infant pairs	IG: trained CHWs vs. CG: standard care	↑				↕
Shamim (2017) [63] ^‡^ Mothers of infants < 6 m	IG1: trained TBAs/CVs vs. CG: TBAs/CVs without special training	↕		↕		
	IG2: trained + supervised TBAs/CVs vs. CG: TBAs/CVs without special training	↕		↕		
**Promotion interventions**						
Cangol (2017) [47] Pregnant—applied to pregnancy preparation course	IG: BF motivation programme based on Pender’s Health Promotion Model—4 times-antenatal period, 1st postnatal day, 4th–6th postnatal week and 4th postnatal month vs. CG: standard care	↑	↔			↔
Gu (2016) [48] Primiparous women companied by husband/mother	IG: Theory of Planned Behaviour (TPB) based intervention programme—individual instruction, group education and telephone counselling vs. CG: routine nursing care	↑				
Moudi (2016) [49] ^‡^ Primiparous women referred to health centre	IG1: peer support group (4 times) vs. CG: routine care	↑				
	IG2: health care provider’s education (4 training sessions) vs. CG: routine care	↑				
Saljughi (2016) [51] Pregnant women	IG: training on promoting BF self-efficacy at 36th week via role playing vs. CG: routine care		↑			
Akram (1997) [46] Pregnant women	IG: promotion of EBF via health messages vs. CG: no health messages	↑	↑			
Reinsma (2016) [50] Pregnant women and their partners	IG: audio programme (Bobbi Be Best) and discussion guide to promote EBF- entertainment education (EBF) vs. CG: entertainment education (injection safety)		↕			
**Peer support interventions**						
Bich (2017, 2017, 2014) [52,53,54] ^†^ Fathers and their pregnant wives from 7 to 30 w gestation	IG: fathers as supporters—BF education material, counselling services at community health centres, invitation to social events and household visits vs. CG: no intervention to fathers	↑	↑			
Dearden (2002) [55] Mothers of infants < 6 m	IG: mother-to-mother support programme of La Leche League Guatemala-BF counselling by trained counsellor vs. CG: usual care	↕				
**Other interventions**						
Yotebieng (2015) [65] Mother–infant pairs	IG1: BFHI steps 1–9, IG2: BFHI steps 1–10 vs. CG: standard care	↔			↔	
Menon (2016) [64] Infants	IG: BF practices at scale-intensified IPC, MM, CM, and PA vs. CG: standard nutrition counselling and less intensive MM, CM, and PA	↑				
Ochola (2013) [66] ^‡^ Pregnant women (34–36 w) attending antenatal clinic	IG1: home based intensive counselling group (HBIC) vs. CG: standard care	↑				
	IG2: facility based semi-intensive counselling group (FBSIC) vs. CG: standard care	↔				

Symbols: † indicates linked studies (studies with similar data reported in >1 studies); ‡, studies with ≥2 comparisons; ↑, positive effect; ↔, no effect; ↕ mixed effect. Note: Positive effect (green), evidence of uniformly favourable impacts across one or more outcome measures, analytic samples (full sample or subgroups), and/or studies; No effect (red), evidence of uniformly null impacts across one or more outcome measures, analytic samples (full sample or subgroups), and/or studies; mixed effect (orange), evidence of a mix of favourable, null, and/or adverse impacts across one or more outcome measures, analytic samples (full sample or subgroups), and/or studies.. Abbreviations: BASNEF, beliefs, attitudes, subjective norms and enabling factors; BF, breastfeeding; BHFI, baby friendly hospital initiative; CG, control group; CHW, community health worker; CM, community mobilisation; CV, community volunteer; EBF, exclusive breastfeeding; IG, intervention group; IMNCI, integrated management of childhood illness; IPC, interpersonal counselling; LBW, low birth weight; MIYCN, maternal infant and young child nutrition; MM, mass media; PA, policy advocacy; TBA, traditional birth attendant.

**Table 3 nutrients-13-00681-t003:** Synthesis using vote counting based on direction of effects for five intervention categories.

Intervention Categories	Outcomes (Proportion of Effects ^†^, 95% CI, *p* Value)				
	Breast-Feeding Practices	Caregiver K/S/P	Healthcare Staff K/S/P	Morbidity	Anthropo-Metry
Counselling	0.58 (0.31–0.80, *p* = 0.77)	NA	-	0.66 (0.20–0.93, *p* = 1.00)	0.33 (0.06–0.79, *p* = 1.00)
Education	0.80 (0.49–0.94, *p* = 0.10)	0.75 (0.30–0.95, *p* = 0.62)	-	0	NA
Training	0.55 (0.26–0.81, *p* = 1.00)	-	0.33 (0.06–0.79, *p* = 1.00)	NA	0
Promotion	1.00 (0.56–1.00, *p* = 0.06)	0.50 (0.15–0.84, *p* = 1.00)	-	-	NA
Peer Support	NA	NA	-	-	-

^†^ Proportion of effects is calculated as *p* = u/n, where u = number of effects favouring the intervention, and n = number of comparisons. NA, not applicable, ≤2 studies/comparisons; K/S/P, knowledge/skills/practices.

## Data Availability

Not applicable.

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
