# Peer review of "Effectiveness of Breastfeeding Support Packages in Low- and Middle-Income Countries for Infants under Six Months: A Systematic Review"

_nutrients, 2021, doi:10.3390/nu13020681_

Round 1

Reviewer 1 Report

This a big public health study about the well nutrition and well being of <6 months old infant ; It's retrospective study , among 15000 there are finally 41 which this work is retained

ThisThis is not meta analyses but a narrative taking in count the data concerning strategy to encourage the human milk in the poor countries to revert wasting and low weight <6 month.

The data was qualitative data with two investigator for each , one internal and an external .

The conclusion was describe a strategy to increase breastfeeding in the <6 months infants, since in total there was only 37% of breast-fed Infant in this study.

The limitation of this work was no anthropometric data to see the influence on the growth of infants .

The work is important and has good explanation. However there was many abbreviations which slow the lecture.

In the Statistic Value the p of the comparaison were never significatives between 0.06 and 1.

Reviewer 2 Report

The manuscript entitled “Effectiveness of breastfeeding support packages in low- and middle-income countries (LMICs) for infants under six months: A systematic review” presents an important issue, however it requires some corrections.

Title – please remove abbreviation from the title

Abstract:

  • Please remove the words “Background” / “Methods”/ “Results” / “Conclusions” -  The abstract should be a single paragraph and should follow the style of structured abstracts, but without headings.
  • Please add the period of data of databases searching.

Materials and Methods:

  • Line 123-124 – “One reviewer (RR) screened all titles and abstract” – why only one person? The more than 15000 studies was identified – this a huge number/risk of mistake for one person. Please describe in detail how authors reduce this risk.
  • Figure 1. Flow diagram of included and excluded studies (should be presented in this section)
  • Figure 1 is of poor resolution (should be improved)
  • The population is mixed (Pregnant women; Infants, Mothers) – this should be unifed
  • In materials and method section authors stated that “Outcome: Studies reporting on at least one of the following outcomes- breastfeeding practices, knowledge/skills/practices of healthcare staff and/or caregivers.”, whereas in table 2 indicate also “Morbidity and Anthropometry” – it should be corrected. The morbidity and anthropometry should be removed. Moreover, it was not indicated in search strategy.  The discussion should be corrected accordingly.
  • More information about “vote counting” in table 3 should be presented in the materials and method section.
  • The bias should be assessed (e.g. by using the Newcastle–Ottawa Scale (NOS)).

Minor comments:

  • line 385 – “(Moudi et al. 2016)” – please correct the references/cite style
  • Lines 417-418 – “Similar findings were also reported by others [71]. A 2016 review identified healthcare providers as key players in education and encouraging mothers to breastfeed [15].” – these sentences should be corrected. Authors should present information in more detailed and precise way.

Round 2

Reviewer 2 Report

I appreciate the great efforts that the authors have made in response to my questions and concerns. However, one issue is sill unsolved. 

As an outcome the “Morbidity and Anthropometry” were not indicated in search strategy, so some important publications may be missing. The “morbidity and anthropometry” should be removed. The discussion should be corrected accordingly.
